# Molecular Dynamics Simulation of the Effect of Defect Size on Magnetostrictive Properties of Low-Dimensional Iron Thin Films

**DOI:** 10.3390/nano13233009

**Published:** 2023-11-23

**Authors:** Hongwei Yang, Panpan Ma, Meng Zhang, Lianchun Long, Qianqian Yang

**Affiliations:** 1Faculty of Science, Beijing University of Technology, Beijing 100124, China; ppm13021017300@163.com (P.M.); mel98ray99@163.com (M.Z.); 2Faculty of Materials and Manufacturing, Beijing University of Technology, Beijing 100124, China; longlc@bjut.edu.cn

**Keywords:** magnetostriction, molecular dynamics, defect size, low dimension, iron thin film

## Abstract

Defects are an inevitable occurrence during the manufacturing and use of ferromagnetic materials, making it crucial to study the microscopic mechanism of magnetostrictive properties of ferromagnetic materials with defects. This paper conducts molecular dynamics simulations on low-dimensional iron thin films containing hole or crack defects, analyzes and compares the impact of defect size on magnetostrictive properties, and investigates the microscopic mechanism of their effects. The results indicate that the saturation magnetostrictive strains of the defect models do not increase monotonically as the defect size increases. Additionally, it is discovered that the arrangement of atomic magnetic moments in the initial magnetic moment configuration also affects the magnetostrictive properties. When controlling the size of the hole or crack within a certain defect area, it is found that the hole size has less influence on the initial magnetic moment configuration, resulting in a smaller corresponding change in the saturation strain and thus having a lesser impact on the magnetostrictive properties. Conversely, when the crack size changes, the arrangement of the atomic magnetic moments in the initial magnetic moment configuration changes more significantly, resulting in a greater corresponding change in saturation strain, and thus having a greater impact on the magnetostriction performance.

## 1. Introduction

Magnetostriction refers to the alteration in length or volume of ferromagnetic materials when they are magnetized in an external magnetic field [1,2,3,4,5]. Materials with excellent magnetostrictive properties hold significant potential for various applications including sensors, brakes, micro-electro-mechanical systems, energy harvesting, and underwater sonar scanning [6,7,8,9,10]. Magnetostrictive materials, including Fe [11,12,13,14,15], Terfenol-D (terbium dysprosium iron alloy) [16], and Fe-Ga alloy [17,18], have been widely studied.

Theoretical simulation contributes to the understanding of the microscopic mechanism of the magnetostrictive properties, offering the potential to synthesize nanomaterials with ideal magnetostrictive properties. Wang et al. simulated the substitution of a small quantity of Cu for Ga in a particular configuration, resulting in a twofold increase in the magnetostriction of Fe-Ga alloy [19]. Matyunina et al. theoretically studied how the atomic arrangement and tetragonal distortion impacted the magneto-crystalline anisotropy properties and magnetostrictive behavior of Fe-Ga alloy [20].

However, the theoretical study of magnetostriction pays less attention to the defects, while defects are inevitably formed during the manufacturing and application of ferromagnetic materials. Studying the dependence of magnetostriction on defects is vital for improving magnetostrictive performance. We have investigated the effect of the spacing of defects on the magnetostrictive behavior and magnetic moment evolution of iron thin films through a molecular dynamics approach [21]. But defect size is also a crucial factor influencing magnetostrictive properties. There are very few papers, especially from a theoretical and microscopic perspective, that explore the impact of defect size on the magnetostrictive properties of materials.

So far, most of the magnetostrictive materials are based on iron. Therefore, in this letter, we investigate the impact of defect size on the magnetostrictive properties of 2D iron thin films, using molecular dynamics simulations. Our study focuses on two types of defects: crack defects and hole defects. Specifically, our goal is to construct a link between macroscopic magnetostrictive behavior and microscopic atomic magnetic moments and to reveal the microscopic mechanism of action.

## 2. Computational Model

In this paper, we simulate the magnetostriction process of body-centered cubic (bcc) iron thin films containing various crack lengths or hole sizes using molecular dynamics. We utilize the LAMMPS (Large-scale Atomic/Molecular Massively Parallel Simulator) software to perform the simulations, and the results are visualized using the Open Visualization Tool (OVITO) software.

The geometric representation of the produced thin film can be observed in Figure 1. The model is oriented along three axes: the *x*-axis along [100], the *y*-axis along [010], and the *z*-axis along [001]. The dimensions of the model are 160*a* × 160*a* × *a* (457.6 Å × 457.6 Å × 2.86 Å), where “*a*” denotes the lattice constant of bcc iron, which has a value of 2.86 Å. The model contains 51,521 iron atoms. The defect model is created by removing atoms from the center region of the model, where *N_x_* and *N_y_* represent the number of atoms removed along the *x* and *y* directions, respectively. For hole defects shown in Figure 1a, both *N_x_* and *N_y_* have equal values. Specifically, *N_x_* and *N_y_* take on the values of 2, 4, 6, 8, 10, 12, and 14, resulting in a defect area size of (*N_x_* × *a*) × (*N_y_* × *a*). This means that the area ranges from 4*a*^2^ to 196*a*^2^. In the case of transverse crack defects, as seen in Figure 1b, *N_x_* takes on the values of 4, 10, 20, 40, 60, 80, and 100. The area size is (*N_x_* × *a*) × 2*a*, resulting in an area range of 8*a*^2^ to 200*a*^2^. Similarly, for vertical crack defects like those shown in Figure 1c, *N_y_* takes on the values of 4, 10, 20, 40, 60, 80, and 100. The area size is 2*a* × (*N_y_* × *a*), and the area also ranges from 8*a*^2^ to 200*a*^2^.

## 3. Calculation Method

To precisely depict the interactions between atoms, the simulation employs various potential functions such as the EAM potential (embedded atom method) [22], spin/exchange potential [23,24,25], and spin/Néel potential [23].

The EAM potential is primarily utilized in molecular dynamics simulations of metals, which characterizes the total potential energy of the system through the following expression:(1)$$E_{T}=\sum_{i}F_{i}(\rho )+\frac{1}{2}\sum_{i,j(j\neq i)}V(r_{ij})$$
(2)$$\rho =\sum_{i\neq j}f_{i}(r_{ij})$$
where *F_i_* represents the embedding energy of an individual atom *i* within a background electron cloud of density *ρ*, *V* represents the interaction potential, and *r_ij_* refers to the distance between atom *i* and *j*. In this paper, the embedding atom potential of the iron is utilized. 

In ferromagnetic materials, spontaneous magnetization mainly arises from the exchange interactions between spins, so the spin/exchange potential is considered. It is used to calculate the exchange interactions between spins, and the strength of the exchange interactions can be defined by the following function [23,24,25]:(3)$$J\left(r_{ij}\right)=4\alpha {\left(\frac{r_{ij}}{\delta }\right)}^{2}\left(1-\gamma {\left(\frac{r_{ij}}{\delta }\right)}^{2}\right)e^{-{\left(\frac{r_{ij}}{\delta }\right)}^{2}}\Theta \left(R_{c}-r_{ij}\right)$$
where *R_c_* is the rigid cutoff radius, and *α*, *δ*, and *γ* are the three coefficients of the function. In this paper, the specific parameter values taken were, respectively, *R_c_* = 4.0 Å, *α* = 0.1 eV, *δ* = 1.841 (dimensionless), and *γ* = 0.2171 eV. The exchange interaction energy between spins can be obtained from
(4)$$H_{ex}=-\sum_{i,j}^{N}J_{ij}\left(r_{ij}\right){\;s}_{i}\cdot s_{j}$$
where ***s****_i_* and ***s****_j_* are two unit vectors representing the magnetic spins of two particles, and *N* is the total number of particles.

Magnetocrystalline anisotropy is the main cause of magnetostriction. The spin/Néel potential serves as a description of the two-site magnetic anisotropy, defined as [23]
(5)$$H_{Néel}=-\sum_{i,j=1,i\neq j}^{N}g_{1}\left(r_{ij}\right)\left(\left(e_{ij}\cdot s_{i}\right)\left(e_{ij}\cdot s_{j}\right)-\frac{1}{3}s_{i}\cdot s_{j}\right)$$
where $e_{ij}=\frac{r_{i}-r_{j}}{\left|r_{i}-r_{j}\right|}$ is the normalized separation vector, and the function $g_{1}\left(r_{ij}\right)$ is the same as the Bethe–Slater function used to fit the exchange interaction (see Equation (3)).

The boundary conditions of the model are set to SSP, which stands for non-periodic and shrink-wrapped in the *x* and *y* directions and periodic in the *z* direction. The atomic magnetic moment of the iron atom is 2.2 *μ*_B_ (Bohr magneton), and the magnetic moments of the atoms are initially set to a random distribution to ensure an overall magnetization intensity close to zero. The temperature is set to 300 K. All the simulations are conducted for a canonical (NVT) ensemble with the Langevin thermostat. The time step is 5 fs, and the data are output every 50 fs. The system is relaxed using the above method so that the system is in equilibrium and the initial configuration of the atoms can reach a stable state. The magnetostrictive strain is simulated by applying different magnitudes of magnetic fields along the positive direction of the *x*-axis to the initial model after relaxation equilibrium, and the simulation process is performed using the LAMMPS molecular dynamics simulation software.

## 4. Results and Discussion

### 4.1. The Effect of the Size of the Hole on Magnetostrictive Properties of Thin Films

Before examining the impact of hole size, the defect-free model is analyzed using the aforementioned calculation method. Figure 2 displays the initial magnetic moment configuration diagram of the defect-free model, along with magnetization configuration diagrams under various magnetic fields. Each atomic magnetic moment is projected onto the *xy*-plane to produce these configuration diagrams, with the direction of the arrow in the diagram indicating the direction of the atomic magnetic moment in its respective region. The arrow size, however, holds no significance.

In Figure 2a, a magnetization vortex in the counterclockwise direction appears in the defect-free initial model without an applied magnetic field, located in the upper part of the model center. The atomic magnetic moment in the center of the vortex along the *z*-axis direction exhibits no deflection in the *x* and *y* directions, resulting in a blank projection in the *xy* plane. Subsequently, a magnetic field of 0.375 *H*_m_ (*H*_m_ is the applied external magnetic field when the model reaches saturation) is applied in the positive *x*-axis direction to the defect-free initial model, as illustrated in Figure 2b, causing the atomic magnetic moment in most of the model to align with the magnetization direction. When the magnetic field is further increased to *H*_m_, as depicted in Figure 2c, only the atomic magnetic moments at the left and right boundaries of the model fail to turn toward the positive *x*-axis direction.

The case with hole defects is studied as follows. We provide initial magnetic moment configuration diagrams for models containing different hole sizes in Figure 3. Specifically, Figure 3a,b show the initial magnetic moment configuration diagrams of the 2*a* × 2*a* and 4*a* × 4*a* models, which are similar to the defect-free model except for the hole defect in the center replacing the magnetization vortex of the defect-free model. The overall magnetization intensity is about zero. When the defect increases to 6*a* × 6*a*, some changes in the initial magnetic moment configuration of the model can be seen in Figure 3c, which is attributed to the fact that the surface stress around the defect induces atomic reconstruction as the size of the defect increases, leading to a change in the initial configuration. In Figure 3c–g, we observe that the initial magnetic moment configuration diagrams are all similar, with the majority of atomic magnetic moments following the same transition direction in each model, and the magnetic moments of the atoms located near the boundary are oriented parallel to it. The difference is that the magnetic moments of most of the atoms in the 6*a* × 6*a* model (Figure 3c) and the 10*a* × 10*a* model (Figure 3e) are oriented at an angle of less than 90 degrees from the positive *x*-axis direction, while the magnetic moments of atoms in the 8*a* × 8*a* model (Figure 3d), the 12*a* × 12*a* model (Figure 3f), and the 14*a* × 14*a* model (Figure 3g) are oriented at an angle of more than 270 degrees from the positive *x*-axis direction.

Utilizing the aforementioned configurations as the initial models, varying magnitudes of magnetic fields are applied in the positive direction of the *x*-axis, and the magnetostrictive strain fitting curves are generated under differing magnetic fields, as depicted in Figure 4. The magnetostrictive strain curve of the defect-free model is added for comparison. The vertical axis represents the magnetostrictive strain, while the horizontal axis is the normalized magnetic field strength. *H* represents the size of the applied magnetic field, and *H*_m_ signifies the saturation magnetic field.

As depicted in Figure 4, the magnetostrictive strain in the *x*-direction increases with the increase in the applied magnetic field for both the hole defect model and the defect-free model. When the strain value of a model becomes stable, it is deemed to have reached saturation under the *H*_m_ magnetic field. The degree of overlap between each curve in the figure reveals that the size of the hole defect has a relatively minor effect on the magnetostrictive strain within certain limits.

To investigate the microscopic mechanism of the magnetostrictive behavior, the impact of the defect size on the arrangement of atomic magnetic moments in the model under a saturation magnetic field is further studied. The diagrams in Figure 5 depict the magnetization configurations of hole defect models under the *H*_m_ magnetic field. As shown in the figure, the magnetic moments of most of the atoms at the left and right boundaries of the model fail to turn toward the magnetization direction. Inside the model, the atomic magnetic moments are essentially along the positive *x*-axis direction. However, when the side length increases to 14*a*, the atomic magnetic moments around the defect no longer completely turn toward the positive *x*-axis direction (see Figure 5g).

### 4.2. The Effect of the Size of the Transverse Crack on Magnetostrictive Properties of Thin Films

Figure 6 presents the magnetostrictive strain fitting plots for various transverse crack models. Similar to the hole model, the magnetostrictive strain in the *x*-direction of the transverse crack defect model exhibits a positive correlation with the magnitude of the magnetic field. In the case of the 80*a* × 2*a* model and the 100*a* × 2*a* model, a contraction trend with negative strain values in the *x*-direction is observed when the applied magnetic field is small. Additionally, the degree of overlap among the curves in the figure suggests that the size of the transverse crack has a more substantial impact on the magnetostrictive strain compared to the hole size.

Figure 7 depicts the initial magnetic moment configurations of various transverse crack defect models. From Figure 7a–c, it can be observed that in the initial models of 4*a* × 2*a*, 10*a* × 2*a*, and 20*a* × 2*a*, atomic magnetic moments oriented in the transition direction occupy the major part, the atomic magnetic moments located at the upper and lower boundaries are oriented close to the positive direction of the *x*-axis, and those at the left and right boundaries are along the positive direction of the *y*-axis. When the defect increases to 40*a* × 2*a*, the initial magnetic moment configuration of the model begins to change significantly, as shown in Figure 7d; the atomic magnetic moments above and below the defect are along the positive direction of the *x*-axis; and those at the left and right boundaries are oriented in the transition direction. In comparison to the other six initial magnetic moment configurations of transverse cracks displayed in Figure 7, this model has the most atoms with magnetic moments along the positive direction of the *x*-axis. In the 60*a* × 2*a* model shown in Figure 7e, the magnetic moments of the atoms above the defect are along the negative *x*-axis, while those below are along the positive *x*-axis. The atomic magnetic moments of the entire model are arranged counterclockwise with the defect serving as the center, and the overall magnetization intensity of the model is about 0. Within the 80*a* × 2*a* initial model of Figure 7f, a magnetization vortex with magnetic moments rotating counterclockwise appears, and the magnetic moments of the atoms above the defects are along the *x*-axis negative direction. The magnetic moments of the atoms in the 100*a* × 2*a* initial magnetic moment configuration of Figure 7g are mostly along the negative *x*-axis or transition direction. Since there are more atoms whose magnetic moments are along the negative direction of the *x*-axis in Figure 7f,g, when the magnetic field is applied along the positive direction of the *x*-axis, the magnetic moments of the atoms oriented along the negative direction of the *x*-axis will first turn toward the *y*-axis direction during the process of turning toward the positive direction of the *x*-axis. Hence, negative strain values appear in these two models, as shown in Figure 6. It can be seen that the arrangement of the atomic magnetic moments in the initial model significantly varies with different transverse crack sizes, implying that the size of the transverse crack has a substantial impact on the magnetostrictive properties.

In Figure 8, we can observe the magnetization configurations of the transverse crack models under an *H*_m_ magnetic field. Given that the transverse crack is parallel to the magnetization direction, the atomic magnetic moments surrounding the defect are effortlessly converted to the magnetization direction. As with the hole model, there are also a lot of atomic magnetic moments in the transition state at the left and right boundaries under the *H*_m_ magnetic field.

### 4.3. The Effect of the Size of the Vertical Crack on Magnetostrictive Properties of Thin Films

The magnetostrictive strain fitting curves for the vertical crack models are presented in Figure 9. As observed in the figure, the magnetostrictive strain also increases with the applied magnetic field. When the applied magnetic field is small, the 2*a* × 80*a* model and the 2*a* × 100*a* model exhibit a contraction trend with negative strain values in the *x*-direction. Additionally, the extent of intersection between the curves in the figure indicates that the vertical crack size has a greater impact on the magnetostrictive strain compared to the hole size. 

Figure 10 displays the initial magnetic moment configurations of the vertical crack defect models. In the initial models of 2*a* × 4*a*, 2*a* × 10*a*, and 2*a* × 20*a* in Figure 10a–c, most of the atomic magnetic moments in the middle region are along the transition direction, and those at the boundary tend to be parallel to the boundary direction. In the 2*a* × 40*a* model shown in Figure 10d, the atomic magnetic moments around the defect tend to be parallel to the vertical crack. When the length of the defect increases to 60*a*, as shown in Figure 10e, the initial model changes more, with the magnetic moments of the atoms on either side of the defect along the positive *y*-axis and those at the upper and lower boundaries along the *x*-axis. Figure 10f,g show the initial configurations of the 2*a* × 80*a* model and 2*a* × 100*a* model, with the magnetic moments of the atoms on both sides of the defect either oriented along the positive *y*-axis or at a smaller angle to it, and the magnetic moments of the atoms at the upper and lower boundaries oriented at a smaller angle to the positive *x*-axis. Consequently, under the smaller magnetic field, the atomic magnetic moments struggle to convert to the direction of magnetization. This phenomenon reflects in the macroscopic behavior of the material, which exhibits a shrinking trend along the *x*-axis, as illustrated in Figure 9. The arrangement of the atomic magnetic moments in the initial magnetic moment configuration also differs significantly for different vertical crack sizes, as seen in Figure 10. Therefore, the vertical crack size also has a great effect on the magnetostrictive properties.

Figure 11 illustrates the magnetization configurations of the vertical crack models in the *H*_m_ magnetic field. Since the vertical crack is perpendicular to the magnetization direction, it makes it challenging to turn the magnetic moments of the atoms around the defect toward the magnetization direction. As can be seen from the figure, when the length of the vertical crack is more than or equal to 20*a* (as shown in Figure 11c–g), the atoms that are difficult to magnetize appear around the defect and gradually increase with the increment in the defect size.

### 4.4. Analysis and Comparison

To visually compare the impact of defect size on the magnetostrictive performance of thin films, we present the saturation magnetostrictive strain versus defect area plots for the hole defect and crack defect models in Figure 12. The horizontal axis represents the defect area size, and the vertical axis represents the magnetostrictive strain. It is evident from the figure that the saturation strain values for both the hole defect and crack defect models do not monotonically increase with an increase in defect size.

The black line in Figure 12 illustrates the relationship between the saturation strain of the hole defect model and its corresponding defect area, where the side length of the hole varies from 2*a* to 14*a*, i.e., the defect area increases from 4*a*^2^ to 196*a*^2^. The model exhibits the highest saturation magnetostriction strain value of approximately 0.62% when the hole size is 8*a* × 8*a*. Conversely, the model displays the lowest saturation magnetostriction strain value of approximately 0.50% when the hole size is 6*a* × 6*a*. The magnitude of variation in magnetostrictive strain, which is the difference between the maximum and minimum values, is approximately 0.12%.

The red line in Figure 12 displays the saturation strain versus defect area for the transverse crack defect model. The width of the transverse crack is 2*a*, while its length increases from 4*a* to 100*a*, corresponding to a defect area ranging from 8*a*^2^ to 200*a*^2^. Notably, the 40*a* × 2*a* model exhibits a maximum saturation strain of about 0.71%, while the 100*a* × 2*a* model shows a minimum saturation strain of about 0.15%. Overall, the transverse crack models demonstrate a variation in saturation magnetostriction strain of 0.56%. The blue line in Figure 12 pertains to the vertical crack model, with the area also ranging from 8*a*^2^ to 200*a*^2^. Specifically, the 2*a* × 40*a* model boasts a maximum saturation strain of about 0.79%, whereas the 2*a* × 100*a* model has a minimum saturation strain of about 0.35%. The variation in saturation magnetostrictive strain for the vertical crack models amounts to 0.44%.

Comparing the magnitude of change of the saturation strains, it can be obtained that the size of the hole defect has a minor effect on magnetostrictive performance, whereas the size of the crack defect exerts a more significant impact. Furthermore, the effect of transverse crack size is slightly larger than that of vertical crack size. The microscopic mechanism is due to the magnetostrictive properties of the material directly influenced by the arrangement of atomic magnetic moments in the initial magnetic moment configurations. Comparing the impact of hole and crack size on the initial configuration, it can be observed that the hole size has little effect on the initial magnetic moment configuration, and thus has a minor effect on the magnetostrictive properties. In contrast, the change in the crack size results in a considerable change in the arrangement of atomic magnetic moments within the initial magnetic moment configuration. The change in the atomic magnetic moment arrangement for transverse cracks is slightly more significant than that for vertical cracks, resulting in a slightly larger effect of transverse crack size on the magnetostrictive properties than that of vertical crack size.

## 5. Conclusions

This paper investigates the impact of hole and crack defects of varying sizes on the magnetostrictive properties of low-dimensional iron thin films, using molecular dynamics methods. This study examines the mechanism behind the effect of defect size on magnetostrictive behavior, from the perspective of microscopic atomic magnetic moments. The results demonstrate that the size of the defect has a notable impact on the magnetostrictive properties of the thin films. Specifically, the effect of crack size is significant, with transverse cracks having a slightly greater impact than vertical cracks. Conversely, the impact of hole size on magnetostrictive properties is relatively small. This is demonstrated through the following aspects.

The saturation magnetostrictive strain of the defect model does not exhibit monotonic variation with increasing defect size.When the initial magnetic moment configuration contains more atoms with magnetic moments oriented opposite or perpendicular to the magnetization direction, it becomes more difficult to align the magnetic moments of the atoms with the magnetization direction in a small magnetic field. This can lead to a tendency for the sample to shrink in the magnetization direction.By controlling the variation in defect size within a certain defect area interval, the saturation magnetostrictive strain varies by approximately 0.12% for the hole models, nearly 0.56% for the transverse crack models, and about 0.44% for the vertical crack models. By comparing the magnitudes of these changes, it can be concluded that the size of the hole defect has a small effect on the magnetostrictive performance of the thin films, whereas the size of the crack defect has a large effect on the magnetostrictive performance. Additionally, the transverse crack has a slightly larger effect than the vertical crack.At the microscopic level, the magnetostrictive properties are influenced by the arrangement of atomic magnetic moments in the initial magnetic moment configuration. The hole size has little effect on the initial magnetic moment configuration and thus has little effect on the magnetostrictive properties. Changes in the size of cracks, on the other hand, have a significant impact on the arrangement of atomic magnetic moments in the initial magnetic moment configuration. The change in atomic magnetic moment arrangement is slightly greater in the transverse crack model than in the vertical crack model; thus, the effect of transverse crack size on magnetostrictive properties is slightly larger than the effect of vertical crack size on magnetostrictive properties.

## Figures and Tables

**Figure 1 nanomaterials-13-03009-f001:**
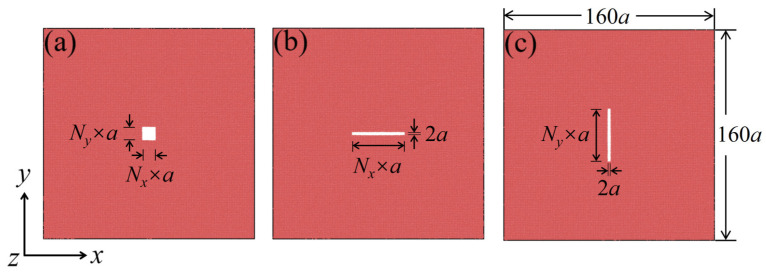
Molecular dynamics models of iron thin films containing different defect shapes: (**a**) holes; (**b**) transverse cracks; (**c**) vertical cracks.

**Figure 2 nanomaterials-13-03009-f002:**
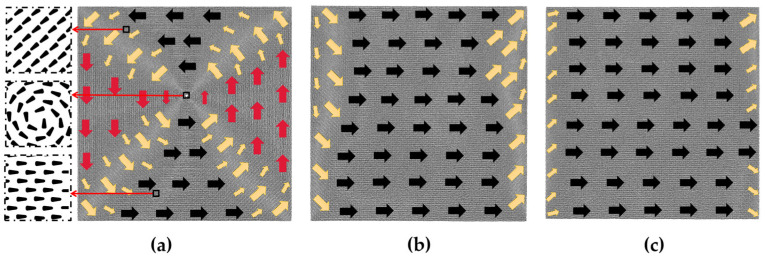
(**a**) Initial magnetic moment configuration diagram of the defect-free model without applied magnetic field. Here, the three diagrams on the left show the atomic magnetic moments that are further zoomed in. The magnetization configuration diagrams of the defect-free model under 0.375 *H*_m_ (**b**) and *H*_m_ (**c**).

**Figure 3 nanomaterials-13-03009-f003:**
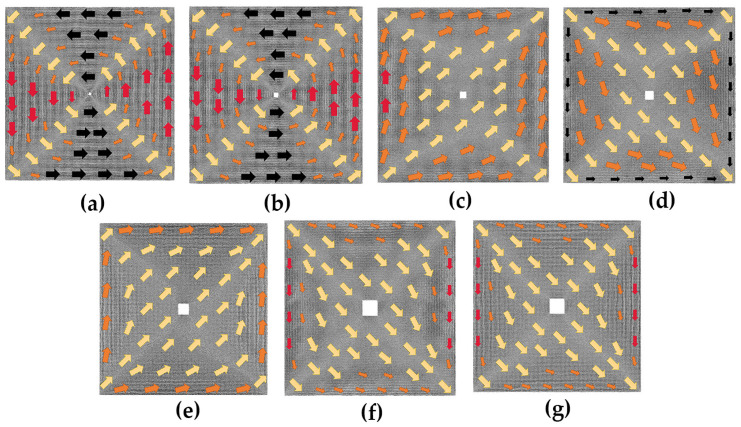
Initial magnetic moment configuration diagrams of the hole defect models without applied magnetic field; the geometric size of the hole defect is in the following order: (**a**) 2*a* × 2*a*; (**b**) 4*a* × 4*a*; (**c**) 6*a* × 6*a*; (**d**) 8*a* × 8*a*; (**e**) 10*a* × 10*a*; (**f**) 12*a* ×12*a*; (**g**) 14*a* × 14*a*.

**Figure 4 nanomaterials-13-03009-f004:**
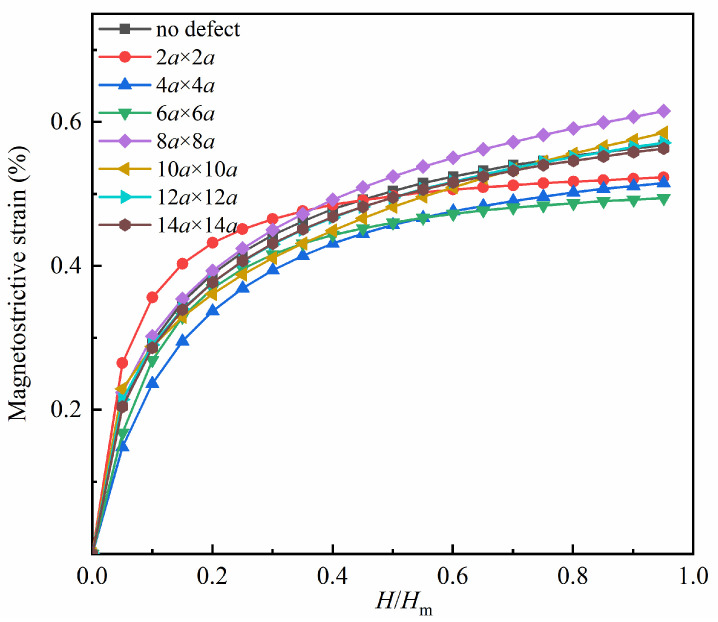
Comparison of magnetostrictive strains in *x*-direction for the hole defect models and the defect-free model.

**Figure 5 nanomaterials-13-03009-f005:**
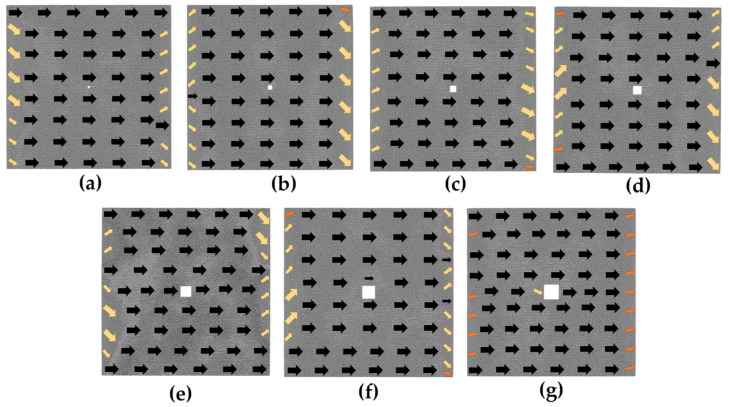
Magnetization configuration diagrams of the hole defect models under 1 *H*_m_ magnetic field; the geometric size of the hole defect is in the following order: (**a**) 2*a* × 2*a*; (**b**) 4*a* × 4*a*; (**c**) 6*a* × 6*a*; (**d**) 8*a* × 8*a*; (**e**) 10*a* × 10*a*; (**f**) 12*a* ×12*a*; (**g**) 14*a* × 14*a*.

**Figure 6 nanomaterials-13-03009-f006:**
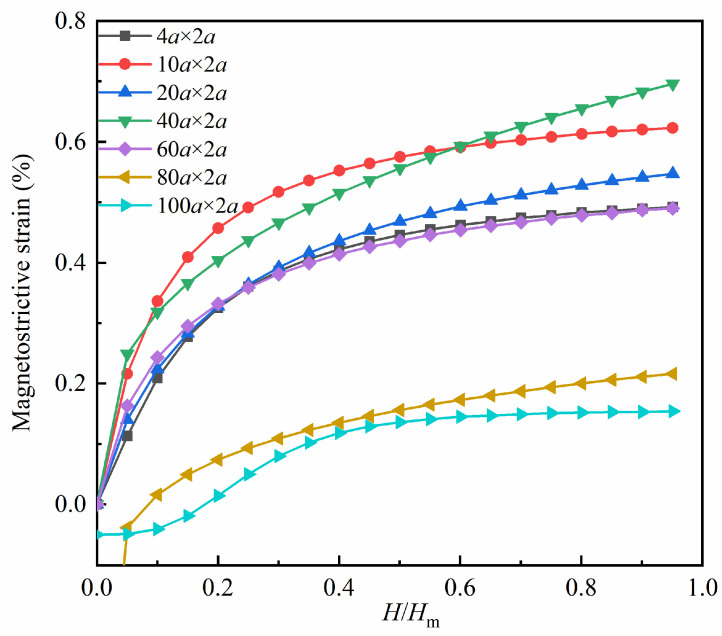
Magnetostrictive strains in the *x*-direction for the transverse crack defect models.

**Figure 7 nanomaterials-13-03009-f007:**
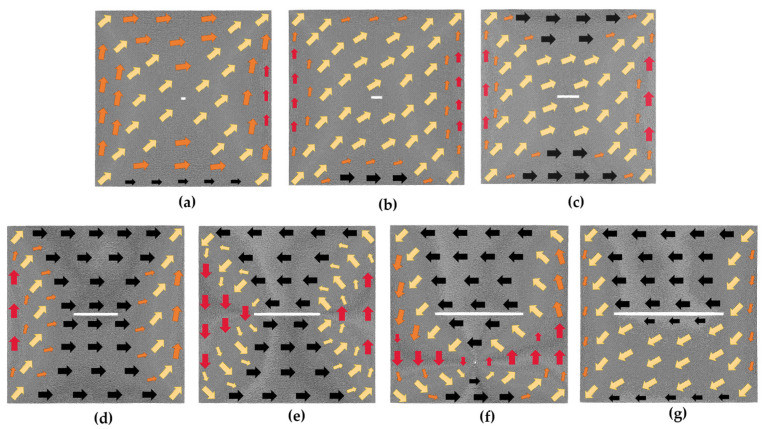
Initial magnetic moment configuration diagrams of the transverse crack defect models without applied magnetic field; the geometric size of the transverse crack is in the following order: (**a**) 4*a* × 2*a*; (**b**) 10*a* × 2*a*; (**c**) 20*a* × 2*a*; (**d**) 40*a* × 2*a*; (**e**) 60*a* × 2*a*; (**f**) 80*a* ×2*a*; (**g**) 100*a* × 2*a*.

**Figure 8 nanomaterials-13-03009-f008:**
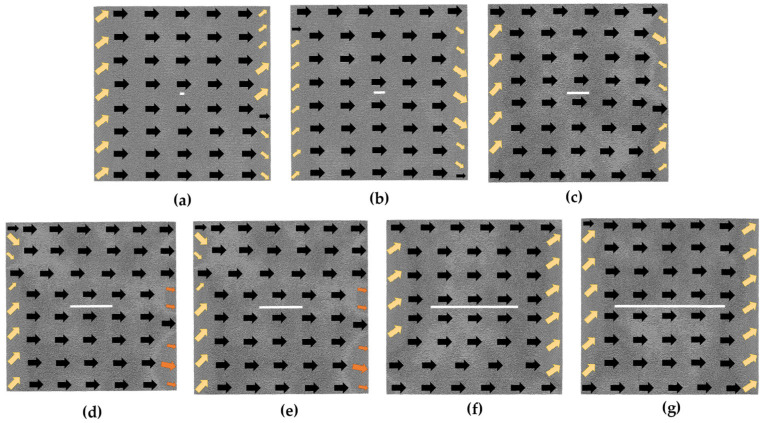
Magnetization configuration diagrams of the transverse crack defect models under 1 *H*_m_ magnetic field; the geometric size of the transverse crack is in the following order: (**a**) 4*a* × 2*a*; (**b**) 10*a* × 2*a*; (**c**) 20*a* × 2*a*; (**d**) 40*a* × 2*a*; (**e**) 60*a* × 2*a*; (**f**) 80*a* ×2*a*; (**g**) 100*a* × 2*a*.

**Figure 9 nanomaterials-13-03009-f009:**
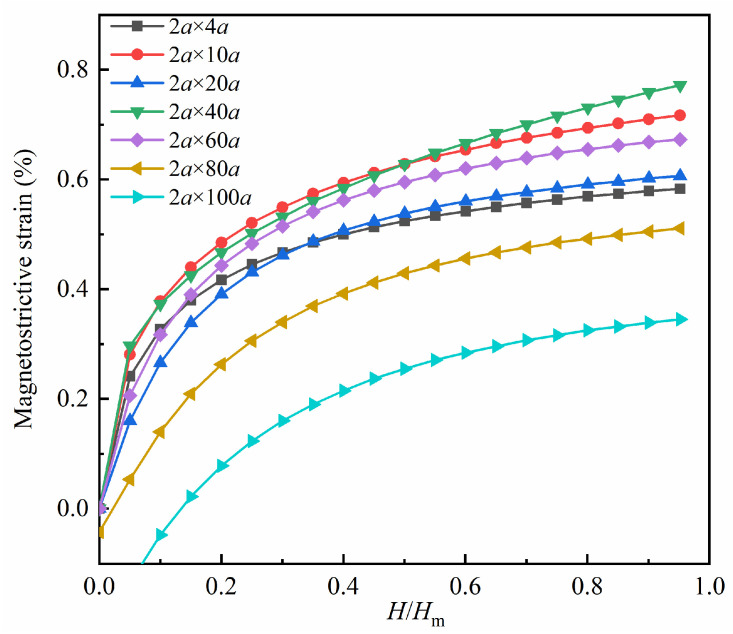
Magnetostrictive strains in the *x*-direction for the vertical crack defect models.

**Figure 10 nanomaterials-13-03009-f010:**
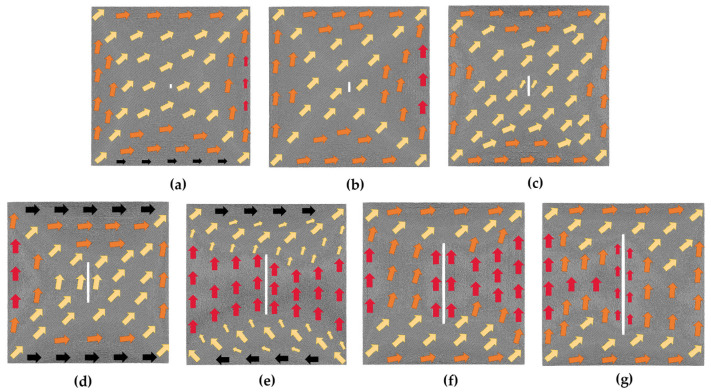
Initial magnetic moment configuration diagrams of the vertical crack defect models without applied magnetic field; the geometric size of the vertical crack is in the following order: (**a**) 2*a* × 4*a*; (**b**) 2*a* × 10*a*; (**c**) 2*a* × 20*a*; (**d**) 2*a* × 40*a*; (**e**) 2*a* × 60*a*; (**f**) 2*a* × 80*a*; (**g**) 2*a* × 100*a*.

**Figure 11 nanomaterials-13-03009-f011:**
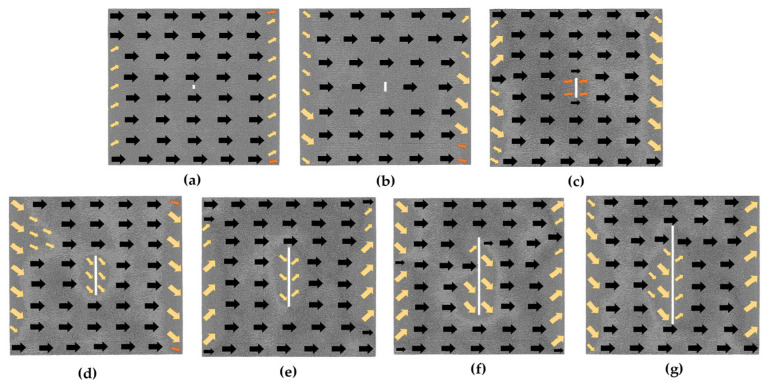
Magnetization configuration diagrams of the vertical crack defect models under 1 *H*_m_ magnetic field; the geometric size of the vertical crack is in the following order: (**a**) 2*a* × 4*a*; (**b**) 2*a* × 10*a*; (**c**) 2*a* × 20*a*; (**d**) 2*a* × 40*a*; (**e**) 2*a* × 60*a*; (**f**) 2*a* × 80*a*; (**g**) 2*a* × 100*a*.

**Figure 12 nanomaterials-13-03009-f012:**
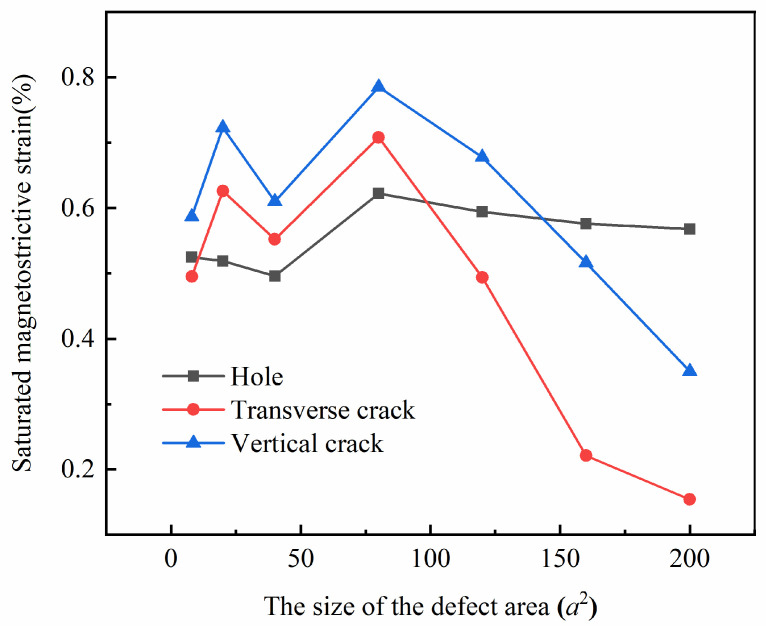
Comparison of saturated magnetostrictive strains.

## Data Availability

The data that support the findings of this study are available from the corresponding author upon reasonable request.

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
