# Peer review of "Molecular Dynamics Simulation of the Effect of Defect Size on Magnetostrictive Properties of Low-Dimensional Iron Thin Films"

_nanomaterials, 2023, doi:10.3390/nano13233009_

Round 1

Reviewer 1 Report (Previous Reviewer 2)

Comments and Suggestions for Authors

Work was revised in accordance with requests of reviewers and can be published in the present state.

Author Response

I would like to express my sincere appreciation for your valuable input and support throughout the review process

Reviewer 2 Report (New Reviewer)

Comments and Suggestions for Authors

The presented manuscript deals with a very interesting issue of modeling the phenomenon of magnetostriction in thin iron films. The presented simulation results were obtained using the LAMMPS (Large-scale Atomic/Molecular Massively Parallel Simulator) package. Since the package works correctly when the boundary conditions of the problem are set correctly, we get correct results. However, I have some reservations about this manuscript.

(1) The manuscript is very similar to the authors' paper written a year earlier, Nanomaterials 2022, 12, 1236, except that the current version deals with the effect of defect size on magnetostriction properties instead of the effect on these properties of the distance between defects. As in the mentioned publication, the manuscript presented for review deals with a purely local system behavior, where physical properties only in the vicinity of a defect are studied. Thus, there is no qualitative progress. There is no discussion of the possible evolution of defects, the formation of new defects, their coalescence, etc. as one would expect.

(2) How the relaxation times for the different types of degrees of freedom, magnetic and non-magnetic, were incorporated into the simulation?

(3) There is no description of the modeled processes, which processes, and how they are modeled using the aforementioned simulation package. On page 3, the authors wrote that simulations were done for a microcanonical ensemble, but at the same time, they wrote about a Langevin thermostat.

(4) The authors should define thermal noise, and what it refers to.

(5) Page 3, In relation to Eq. (3), the authors wrote: ”exchange interactions between spin pairs …”. This is not true. The same error is repeated in the text. Similarly, on Page 3, they have in Eq. (5) two-site magnetic anisotropy (the so-called N\’{e}el } anisotropy) but they wrote ”the anisotropy between magnetic spin pairs”.

(6) Besides, the name N\’{e}el they write as Neel, and neel.

(7) Page 3, line 89: ”Ferromagnetic” which with capital character?

(8) In Eq. (5): How the authors make the fit for the g1 function in their model? The example can be found in part A1 of the article they cite, namely "Journal of Computational Physics 372 (2018) 406–425. A detailed explanation is required for the model in the manuscript.

This form of the manuscript is not suitable for publication.

Author Response

We appreciate the time and effort that you dedicated to providing feedback on our letter. We have studied all of the comments carefully and have made revision which we hope meet with approval. Once again, thank you very much!

Round 2

Reviewer 2 Report (New Reviewer)

Comments and Suggestions for Authors

The revised manuscript meets the conditions for publication.

This manuscript is a resubmission of an earlier submission. The following is a list of the peer review reports and author responses from that submission.

Round 1

Reviewer 1 Report

Comments and Suggestions for Authors

In their contribution “Molecular Dynamics Simulation of the Effect of Defect Size on Magnetostrictive Properties of Low-Dimensional Iron Thin Films“, Yang et al. report on the results of their explorations based on MD simulations. Within the framework of their research, the authors explore the influence of defects in iron thin films on the respective magnetic properties. Although the contents of this remarkable contribution appear to be suited for Nanomaterials, yet, there are certain issues, which should be solved prior to a publication of this work:

- page 1: “The first-principles approach is highly accurate, however, it can only be computed for systems containing up to 100+ atoms [17].” In their contribution, the authors refer to a report published in 2008; however, the limitations of first-principles-based approaches have drastically changed in the meantime because of advanced computational resources. Therefore, I recommend to revise that statement.

- The resolution of the figures is not of utmost quality. For instance, it is hard to identify the starting models shown in the figure 1. Accordingly, I suggest to improve the quality of the diverse figures.

- All simulations were carried out at a constant temperature; however, the magnetic properties of a given material strongly depend on the respective temperature. Hence, do the authors have any idea regarding the influence of temperature on the computed magnetic properties of the iron thin films?

- While the results of the computations are described in detail, there is no sufficient explanation of the computed magnetic properties at the atomic scale. For instance, there is an evident difference between the magnetic properties of bulk materials and those of nanomaterials. Accordingly, is it possible that the computed results are exclusively related to the specific nature of the thin layers? Maybe, first-principles-based approaches for specific sections of the inspected layers could provide some insight into the nature of the respective magnetic interactions.

Comments on the Quality of English Language

Please see the Section Quality of English Language.

Reviewer 2 Report

Comments and Suggestions for Authors

Submitted manuscript describes the attempt to apply molecular dynamics simulations for understanding  low-dimensional iron thin films containing hole or crack defects analyzing and comparing the impact of defect size on magnetostrictive properties. Despite the fact that the subject might be interesting the manuscript causes very confusing impressions. To start with, there are fundamental and very well-known sources related to magnetostriction phenomena (Étienne du Trémolet de Lacheisserie, D.Gignoux, Michel Schlenker, Magnetism: Materials and Applications, Grenoble sciences, Grenoble, Springer 2005, p. 213; R. C. O´Handley, Modern Magnetic Materials (John Wiley & Sons, New York, 1972) p. 740, and others).

Authors make an effort to study thin iron films. However, the majority of the references are related to different magnetostrictive materials and even composites. There is extensive literature related to thin iron films and understanding of their magnetostrictive properties in a view of polycrystalline structure, texture and preparation conditions (Wastlbauer, G. & Bland, J. A. C. Structural and magnetic properties of ultrathin epitaxial Fe films on GaAs(001) and related semiconductor substrates. Advances in Physics 54, 137 (2005); Chlenova et al. Solid State Phenomena Vols 233-234 (2015) pp 657-661 Surface Modification of Thin iron Films in Aromatic Solvents at Ambient Conditions doi:10.4028/www.scientific.net/SSP.233-234.657).

In addition, following paper recently published in the same journal covers the main points of present research. Means that submitted manuscript is just an extension of the work Nanomaterials (Basel), 2022 Apr 6;12(7):1236. doi: 10.3390/nano12071236. Effect of Crack Defects on Magnetostriction and Magnetic Moment Evolution of Iron Thin Films by Hongwei Yang, Meng Zhang , Lianchun Long . Novelty is insufficient for publication.

Comments on the Quality of English Language

Minor proof-read is necessary.